# On Structural and Molecular Order in Cellulose Acetate Butyrate Films

**DOI:** 10.3390/polym15092205

**Published:** 2023-05-06

**Authors:** Malin Nejström, Bo Andreasson, Johanna Sjölund, Alireza Eivazi, Ida Svanedal, Håkan Edlund, Magnus Norgren

**Affiliations:** 1FSCN, Surface and Colloid Engineering, Mid Sweden University, 85170 Sundsvall, Sweden; 2Nouryon, 85467 Sundsvall, Sweden; 3FibRe-Centre for Lignocellulose-Based Thermoplastics, Fibre and Polymer Technology, KTH Royal Institute of Technology, 10044 Stockholm, Sweden

**Keywords:** cellulose acetate butyrate, cholesteric ordering, TOPEM DSC, crystallinity, film, commercial

## Abstract

Cellulose acetate butyrate (CAB) is a possible candidate, being a raw material derived from renewable resources, to replace fossil-based materials. This is due to its thermoplastic properties and the relative ease with which it could be implemented within the existing industry. With a significant amount of variation in CAB on the market today, a knowledge gap has been identified regarding the understanding of the polymer structural arrangement in films. This relates to the underlying mechanisms that regulate CAB film material properties, insights that are important in product development. In this study, commercially available CAB was investigated with XRD, SEM, AFM, and TOPEM DSC in order to obtain physicochemical information related to its micro-structural features in solvent-cast films. The film-forming ability relates mostly to the number of hydroxyl groups, and the semi-crystallinity of the films depends on the type and position of the side groups along the cellulose backbone. The appearance of signs of possible cholesteric ordering in the films could be connected to higher amounts of hydroxyl groups along the backbone that disturb the helix arrangement, while the overall order was primarily related to the butyrate substitution and secondarily related to the molecular weight of the particular CAB studied. Cold crystallization was also observed in one CAB sample.

## 1. Introduction

Biobased plastic, originating from cellulose raw materials, is identified by the European Commission as a possible solution to replace oil-based plastics. Materials with properties similar to those of traditional plastics, but with a lower carbon footprint, are urgently needed [1]. Being an abundant, naturally occurring polymer with interesting properties [2], cellulose constitutes a promising source for the development of more sustainable thermoplastic materials. Unfortunately, in the native form of cellulose, some important physicochemical properties of plastics are absent—for example, the glass transition state, T_g_, is so high that it is of no practical use [3]. Therefore, cellulose as a raw material cannot provide the fast-paced change that is needed. On the other hand, several cellulose derivatives exhibit T_g_ at reasonable temperatures and could be useful as replacements for oil-based plastics [4,5,6].

Cellulose acetate butyrate (CAB) is a cellulose ester, where the hydroxyl groups attached to the glycoside rings in cellulose are substituted, partially or fully, by acetyl and butyryl groups [7]. The ratio between acetyl and butyryl substitution can be varied in their synthesis, and somewhat different properties can be obtained [8]. The total degree of substitution (DS) is regarded as a maximum of three; however, for a degree of polymerization (DP) of less than 134, a slightly higher DS can occur due to the end-group contribution [9]. Both the DS and the molecular weight of cellulose esters can be altered, and both are reported to lead to a variation in material properties [10,11].

CAB is often investigated together with cellulose acetate (CA) and cellulose acetate propionate (CAP), two other cellulose ester derivatives. Several studies on the comparison between these cellulose derivatives have been reported [7,12,13,14,15]. CA is industrially produced on a large scale with several manufacturers, such as Eastman, Cerdia, Daicel, and Celanese; currently, Eastman is among the few who provide CAB and CAP in larger quantities. However, CAB is an important polymer in the bioplastic puzzle, as T_g_ is set at the very broad range of 85–160 °C depending on type [4]. In addition, CAB is easy to dissolve in most bulk solvents, cast films are transparent, and the material is resistant to water [16,17]. The structure and order of CAB have previously been studied with XRD [13,15,18,19,20,21,22,23,24]; however, results from a full set of commercially available CAB materials have not been compiled before, which is invaluable information of a raw material for scalable product development.

Thin films made from CAB are speculated to show signs of cholesteric ordering, where nematic-phase layers are incrementally displaced to produce helical stacking. However, most of the research in these systems is aimed towards how and why this occurs and less towards what possible effect this can inflict upon the film material properties. The reported cholesteric ordering occurring in CAB films has been solely observed with TEM at high butyrate contents and at lower amounts of acetyl groups, together with moderately long polymer chains of approximately 50 to 70 kDa [25,26]. For another butyrate containing cellulose derivative, butyric esters of (2-hydroxypropyl) cellulose (BuPC), it is shown that the pitch height is inversely proportionate to the degree of substitution [27].

A new property concept related to CAB material was proposed recently by Tsioptsias: the glass chemical transition [19,28]). Tsioptsias suggests that the re-crystallization of CAB at elevated temperatures, identified previously by others [24,29,30], is in fact a degradation-induced softening, where the degradation of butyrate side groups increases the overall hydrogen bonding. This study was performed on one type of CAB only and, in that study, it was proposed that other variations in CAB would show the same phenomenon [19]. Regardless of the quality of that particular study, this suggests that there are novel effects to explore within this material and that a study with a broader sample base in terms of variation may provide important information.

In the phase of transitioning from a fossil-based raw material to using CAB in an industrial process, the development progression would be greatly enhanced by readily available information on how and why the material properties will be affected by the polymer structure. The initiating idea for this investigation stems from an initial experiment where two different types of CAB were dissolved and solvent-cast, and the outcome from visual inspection, as well as the tactile properties of the macro films experienced, were surprisingly different from what the variation in butyryl and acetyl group content, and molecular weight differences, might have suggested. As these are the main variations that are generally discussed for CAB, further exploration of the variations occurring in several different CAB samples would be meaningful to fill the current knowledge gap regarding their film formability. The approach for this study was to explore solvent-cast CAB films from several angles at once in order to examine the material in a larger context, and to account for the complexity of the material and the likelihood that many of the properties are linked.

Commercially available CAB was dissolved in acetone, as dense films can be obtained when CAB is cast in that solvent [16]. The films were studied with SEM, XRD, and TOPEM DSC to investigate the underlying mechanism behind the differences in structural properties at different scales between the CAB types.

## 2. Materials and Methods

### 2.1. Materials

CAB with a wide range of structural variations was acquired from Eastman (Kingsport, TN, USA), see the supplier data in Table 1. The naming convention appears to follow CAB-XXY-ZZ, where X is the butyryl content and Z is similar to the viscosity. The acetone used for dissolution, with a purity of >99.8%, was supplied by Sigma-Aldrich (Darmstadt, Germany).

### 2.2. Calculations from the Supplier Data

The supplier data in Table 1 are not representative of how the amount of substitution is conventionally described in academic studies and are recalculated to degree of substitution (DS). The degree of substitution for the acetyl, butyryl, and hydroxyl groups (DS_Ac_, DS_Bu_, and DS_H_, respectively) was calculated using Equation (1), where M_x_ is any of the following: M_H_ = 1.008, M_Ac_ = 43.045, M_Bu_ = 71.099, and M_C6H7O5_ = 159.12; wt%_cellulose_ is the remaining amount when the supplier weight percentage is subtracted, using a corrected value for the hydrogen, calculated using Equation (3). wt%_x_ describes wt%_H_, wt%_Ac_, or wt%_Bu_. The hydroxyl weight percentage given by the supplier includes oxygen as it is, in reality, the OH percentage; therefore, wt%_H_ was calculated using Equation (2), where wt%_OH_ is the value provided by the supplier (Table 1). The total DS was calculated from the weight percentage according to Equation (4). The average molecular weight of one glycose ring unit (M_unit_) was calculated using the DS for each specific sample using Equation (5). The degree of polymerization (DP) was calculated using Equation (6). The equations are based on a conventional number of substance calculations and are formulated in this study.
(1)$${\mathrm{DS}}_{x}=\frac{{\mathrm{wt}\%}_{x}/M_{x}}{{\mathrm{wt}\%}_{\mathrm{cellulose}}/M_{C6H7O5}}$$
(2)$${\mathrm{wt}\%}_{H}=\frac{{\mathrm{wt}\%}_{\mathrm{OH}}{*M}_{H}}{M_{\mathrm{OH}}}$$
(3)$${\mathrm{wt}\%}_{\mathrm{cellulose}}=100-\left({\mathrm{wt}\%}_{\mathrm{Ac}}+{\mathrm{wt}\%}_{\mathrm{Bu}}+{\mathrm{wt}\%}_{H}\right)$$
(4)$${\mathrm{DS}}_{\mathrm{tot}}=\frac{\left({\mathrm{wt}\%}_{H}+{\mathrm{wt}\%}_{\mathrm{Ac}}+{\mathrm{wt}\%}_{\mathrm{Bu}}\right)+/\left(M_{H}+M_{\mathrm{Ac}}+M_{\mathrm{Bu}}\right)}{{\mathrm{wt}\%}_{\mathrm{cellulose}}/M_{C6H7O5}}$$
(5)$$M_{\mathrm{unit}}=M_{C6H7O5}+{\mathrm{DS}}_{H}{*M}_{H}+{\mathrm{DS}}_{\mathrm{Ac}}{*M}_{\mathrm{Ac}}+{\mathrm{DS}}_{\mathrm{Bu}}{*M}_{\mathrm{Bu}}$$
(6)$$\mathrm{DP}=\frac{M_{w}}{M_{\mathrm{unit}}}$$

### 2.3. Molecular Weight Determination by SEC

Size-exclusion chromatography (SEC) was performed using an Agilent 1260 Infinity II (USA) instrument with a polypore 7.8 × 300 mm (2×) column and a pre-column at 35 °C using light scattering. The molecular weight range was 500–2,000,000 Da and the mobile phase was tetrahydrofuran (THF) with a flow rate of 0.6 mL/min and an injection volume of 50 µL. The sample concentration was approximately 1 mg/mL, and sample filtration was carried out with a 0.45 µm polytetrafluoroethylene (PTFE) filter. Data processing was carried out using Astra 7.3.2.19.

### 2.4. Film Preparation

The different cellulose acetate butyrate (CAB) samples were dissolved in acetone to yield 3 wt%. Thereafter, 22.5 g of the solution was carefully poured into a Teflon dish measuring 100 mm in diameter. Then, the dish was placed under a lid with three 5 mm diameter holes that were evenly spaced and positioned in a ventilated cupboard for two days at 12% relative humidity (RH) and a temperature of 24 °C to completely evaporate the acetone. Finally, the film was carefully removed using a small spatula and kept sealed in an airtight bag at room temperature for analysis. Five films of each type of CAB were prepared. A schematic presentation of the film-making process can be found in the Appendix A.

### 2.5. Multi-Frequency Temperature Modulated Differential Scanning Calorimetry (TOPEM DSC)

The differential scanning calorimetry (DSC) measurements were carried out with a METTLER DSC1 instrument (Germany) with the solvent-cast films. All the tests were performed in nitrogen at 50 mL/min. The calibration of the instrument is performed annually by Mettler using indium and zinc for both temperature and heat flow. Before the tests, a check with indium was performed. Samples in the range of 5 to 10 mg were inserted at 25 °C and heated to 260 °C at 2 °C/min using stochastic temperature modulation (TOPEM) at a ± 0.5°C pulse height, with pulse widths in the range of 15 to 30 s. A second scan was performed on the samples after they were cooled ambivalently to room temperature. Aluminum crucibles with pierced lids were used as sample containers, along with an empty crucible as reference.

### 2.6. Atomic Force Microscopy (AFM)

Surface morphology and roughness of the films were analyzed by atomic force microscopy (AFM; Park Systems NX20, Suwon, Republic of Korea). The AFM was operated in contact mode in air. An SD-R30-FM probe (Park Systems, Republic of Korea) with a nominal resonance frequency of 75 kHz and force constant of 2.8 N/m was used. AFM images of 5 × 5 µm^2^ representative areas on the samples were acquired with a scan rate of 0.3–0.6 Hz. The surface roughness parameter, R_q_, was measured from the data using the Park systems XEI 1.8.5 image analysis software. The measurement results on various areas of each sample were almost identical, with less than 10% variations.

### 2.7. X-ray Diffraction (XRD)

X-ray diffraction (XRD) was conducted at room temperature using a Bruker D2 Phaser diffractometer (Germany) with Cu-Kα radiation (wavelength λ = 1.54 Å) at 30 kV and 10 mA, with θ–2θ geometry. The solvent-cast film samples were measured by fixing them onto a silicon single crystal, which was specially cut to provide a low background, free from any interfering diffraction peaks. The samples were rotated at 15 rot/min, providing better averaging over crystallite orientations. The increment was set at 0.02°. The XRD results on the replicate samples of each CAB material were almost identical, with less than 5% variation.

### 2.8. Field Emission Scanning Electron Microscopy (FE-SEM)

The samples for field emission scanning electron microscopy (FE-SEM) were prepared using a Leica UC6 microtome; the solvent-cast film was clamped in an atomic force microscopy holder and cut at room temperature with a diamond histo knife at 5.00 mm/s. All of the samples were coated with a 12 nm layer of carbon. The FE-SEM cross-section images were collected using a Zeiss Sigma 300 and a Zeiss Crossbeam 540 (Germany). Typical parameters during imaging were an accelerating voltage of 2 kV, a working distance of 5 mm, and the use of the secondary electron detector.

### 2.9. Fourier Transform Infrared Spectroscopy (FT-IR)

FT-IR was performed with an FTIR Nicolet Nexus instrument, going from wavenumber 650 cm^−1^ to 4000 cm^−1^ at ambient temperature and 14% RH. It was performed on the film samples with the highest and lowest butyrate content, CAB-551-0.01 and CAB-171-15, respectively. The FT-IR spectra can be found in the Appendix A.

## 3. Results and Discussion

It was deemed useful to obtain and compare the number average (M_n_), the weight average (M_w_), and the Z average (M_z_), as well as the molar-mass dispersity (MMD), previously known as polydispersity index (PDI), all of which were investigated in this study and can be found in Table 2. The degree of substitution (DS), molecular weight of one glucose unit, and the degree of polymerization (DP) were all calculated as average values and can be found in Table 3. The DS values were calculated based on the information supplied by the manufacturer and yielded a total DS with a slight deviation from the expected value of 3.0, which is normal for lower-viscosity CAB [9]. The summarized information in Table 2 and Table 3 indicate that the samples investigated varied significantly in both molecular weight and the type of substitution.

### 3.1. Film Formability Is Related to Molecular Weight and Hydroxyl Groups

Solvent-casting was the technique used to process the raw CAB material into films in this study, and it has also been used in other studies [31,32]. Uncomplicated powder dissolution and film casting influences the processing time and reduces the different operation costs, making it more suited for industrial production. This study used acetone for the solvent-casting process, and the film-forming ability described is based on that method. In this study, the term film formability embraced the homogeneity and visual appearance of the film, the ease with which it was removed from the Teflon dish, and the brittleness of the film during handling and subsequential cutting. In short, this was a relative observation of the films, as compared to each other, and an associated number between one and four was assigned to describe this, with one being poor and four being good. Film formability is a subjective term that describes a set of combined characteristics and, as such, this subjective observation can offer great value in terms of understanding the complexity of the material, as it shows that its practical application is dependent on how these factors combine. When the film formability was compared with the supplier data for the CAB, it was found that the average molecular weight of the polymer was, to a certain degree, correlated. If the molecular weight was similar, the hydroxyl groups in the polymer seemingly affected the film formability as well, as seen in Figure 1.

Regardless of the subjectiveness of the term, it is clear that samples with a high molecular weight and a low number of hydroxyl groups, such as CAB-500-5 or CAB-171-15, had a higher film formability. The opposite is indicated for samples such as CAB-551-0.01, even when the samples were similar in the degree of substitution of more bulky butyrate groups, as was the case for CAB-500-5 and CAB-551-0.01. The effect of chain length is reasoned to be due to the decrease in the degree of freedom (entropy) for longer chains due to entanglements. Longer chains give rise to a decrease in solubility; thus, the solidification of the film occurs faster. The amount of hydroxyl groups will directly affect the hydrogen bonding in the sample, and it is reasonable to assume that less hydrogen bonding will decrease the internal tension of the material, hence the increase in film formability. Additionally, it is known that the molecular weight will have an influence on the solution viscosity; a higher Mn will increase the viscosity [32], and with that, decrease the degree of freedom of the molecules, which gives rise to a film that is easier to handle.

### 3.2. Structural Morphology Investigations

Liquid crystals can be formed in CAB during the evaporation of the solvent when a specific concentration is reached [19,28]. The concentration will vary between CAB types due to the specific molecular structure and chain length, but also within the same sample, since commercial CAB is far from homogeneous over the repeating units and the distribution of chain length. It needs to be better addressed as to how the molecular structure and chain length will affect this in relation to film formability. Industrially, the production of cellulose esters is a heterogeneous process, where undissolved but swollen cellulose is first substituted as much as possible, after which it is de-substituted to reach the targeted product specifications [11]. This means that the hydroxyl groups in the cellulose structure will be over-represented at the more sterically available C6 position, while the substituting groups of acetyl and butyryl will be situated at C2 and C3 to a greater degree. This was examined in detail by Yu and Gray with the reductive cleavage method; of the glucoside units that contained OH groups, 66% and 77% were positioned on the C6 position for CAB-381-20 and CAB-171-15, respectively [33]. The heterogeneous process also means that—as the cellulose was never really dissolved, only swollen—the concentration gradient of substituents in the solution led to an inequal general distribution of the substituent. With three possible substituent positions on each glycose unit and with three different functional substituents, not including the end groups, the number of variants for the “monomer” is 27, although several of them are statistically unlikely. The amount of glycoside units that are completely without a hydroxyl group was 67% and 77% for CAB-381-20 and CAB-171-15, respectively [33]. It is reasonable to assume that the CAB is far from chemically homogeneous, and as it is usually energetically favorable to stack similar substituents together, the film will exhibit domains with enrichment of acetyl and butyryl. Acetyl and butyryl groups are more chemically alike compared with the hydroxyl group, and it is reasonable to assume that they can form crystalline parts. This will influence the order in the films; parts that are rich in acetyl or butyryl, or both, will likely induce crystalline parts, as they are more similar to a homopolymer in those domains. Those parts rich in hydroxyl groups will incorporate irregularities into the system, impede the crystallinity, and thus result a semi-crystalline structure; this is illustrated in Figure 2.

TOPEM DSC is a stochastically modulated DSC, used here to investigate the crystallinity and behavior of the film samples. Modulated DSC is a technique that allows the heat flow to be separated into a kinetic and a specific heat part, allowing the separation of thermal events. The kinetic part is dependent on the heating rate and is also known as the non-reversible part, while the specific heat part is known as the reversible part. With modulated DSC, it is possible to separate events caused by melting (specific heat component, most of total melting); glass transition and heat capacity on the reversible curve; and enthalpic relaxation, evaporation, melting (kinetic component, some of total melting), crystallization, curing, denaturation, and decomposition on the non-reversible curve. The total curve is the sum of the reversible and non-reversible curves and is equivalent to the traditional DSC curve [34,35]. In Figure 3, the modulated DSC curves for the first heating of the CAB films are found. The first heating curves were determined to be of more interest to this paper than the second heating data, as the first contains more information even though some of them are obstructed by the thermal history. The T_g_ found in the second heating data are already found in the literature and are of little interest to explore further [4]. In general, all the samples except CAB-171-15 showed the same behavior, with small variations. In terms of the reversible flow, the small endothermic peak caused by the T_g_ was closely followed by an endothermic melting peak from the specific heat component, almost overlapping each other. The non-reversible curve showed that enthalpic relaxation [36] takes place at the same temperature as the melting and is possibly attributed to melting as well as both are endothermic. The presence of enthalpic relaxation in the samples makes the TOPEM analysis less powerful, as it obscures other, more interesting peaks such as the kinetic component melting and the cold crystallization. Enthalpic relaxation is an endothermic occurrence; as such, the endothermic melting will increase the peak while the exothermic cold crystallization will decrease the size of the enthalpic relaxation. The presence of the T_g_ and a melting peak for all of the samples suggests semi-crystallinity, as was proposed earlier; the crystallinity, however, was not quantifiable from these data, as the melting in the non-reversible curve was obscured by the enthalpic relaxation [37]. The only exception to that in this study was the CAB-171-15 sample, where the cold crystallization likely was large enough to overshadow the enthalpic relaxation, resulting in an exothermic peak on the non-reversible curve. The sample was the same age as the other CAB films and should have similar enthalpic relaxation, leading us to conclude that CAB-171-15 showed a higher level of cold crystallization than the other samples. That suggests that the CAB-171-15 sample expressed higher chain mobility when heated, perhaps amplified by the larger distance between the glass transition and the melting transition compared with the other CAB samples. CAB-171-15 was notable because it had a lower number of butyrate groups and was more similar to a cellulose acetate than any of the others.

Modulated DSC was carried out previously on CAB samples in their raw material powder form [19,28]. It was carried out on the raw material of a sample that is very similar to CAB-381-20, as it is reported to have an M_n_ of 70,000 g/mol, 12–15 wt% acetyl, 35–39 wt% butyryl, and 1.2–2.2 wt% hydroxyl. It is interesting to note that the resulting curves are dissimilar, both in the two referenced studies and in terms of the results of CAB-381-20 from this study. The modulated DSC carried out in the referenced studies was frequency based rather than stochastically derived [19,28].

Figure 4 shows the XRD patterns for the films fabricated by various CAB samples, used to further investigate the semi-crystallinity in the as-prepared samples. In previous work with CAB, discussion on the thermal history resulting from sample preparation and its influence on crystallinity was seldom made [24]. However, it has been shown that with heating over the melting point and slow cooling, the crystallinity of CAB will increase [19]. Hence, it is essential that all samples have experienced the same thermal background for a comparable characterization. The as-prepared films in this study were deemed to have similar thermal backgrounds for comparison as all of them were prepared from a constant weight of 3 wt% CAB solutions, dried at 12% relative humidity (RH) and a temperature of 24 °C for two days, and the XRD results on the replicate samples were almost identical (˂5% variations). The diffractogram peaks for all the CAB films are typical of cellulose tributyrate/acetate butyrate structures [18,38] and have a clear origin from the cellulose diffraction peak [21]. The main diffraction peaks are located at 2Ɵ, about 6.5 and 20°. It has been identified that an increase in the intensity of the first peak (2Ɵ = 6.5°) enhances the crystallinity of the sample. However, the presence of crystallinity has a minor effect on the second peak shape and intensity [38]. As can be observed in Figure 4, the crystalline diffraction peaks in the various CAB types are different, and by dividing the samples into subgroups, where the DS_Bu_ or the molecular weight is similar, some interesting points of observations can be made. For the three samples, CAB-381-0.1, CAB-381-0.5, and CAB-381-2, which all have the exact same DS_Bu_, the intensity of the first peak increased with the decreasing molecular weight. The samples CAB-321-0.1/CAB-381-0.1, CAB-171-15/CAB-381-20, and CAB-500-5/CAB-381-2 are three pairs with similar molecular weights within each pair; in all three cases, the intensity of the first peak increased with increasing DS_Bu_. This suggests that a film made from a CAB polymer with a low molecular weight and high butyrate content is more likely to have a higher crystallinity. This can also be observed in the XRD pattern of a film made from the CAB-551-0.01 sample, which has the lowest molecular weight and highest butyrate content among the CAB samples.

The surface morphology of the CAB films was also evaluated by AFM measurements (Figure 5). The surface roughness (Rq values) of films made from various CAB samples under comparable conditions varies from 7.1 to 24.2 nm. The results revealed that more crystalline and less crystalline CAB films (Figure 4) could be produced with various roughness depending on the target application. For instance, the surface roughness of more crystalline films prepared from CAB-500-5, CAB-381-20, CAB-551-0.01, and CAB-381-0.1 samples were about 7.1, 14.5, 20.5, and 20.8 nm, respectively. On the other hand, the surface roughness of less-crystalline CAB films from CAB 321-0.1, CAB-171-15, CAB-381-0.5, and CAB-381-2 samples can vary between 7.1 and 24.2 nm. The changes in the surface morphology of CAB films can result from various phenomena involved in dissolution and drying conditions for the CAB samples with different characteristics. However, identifying the main influential parameter and its underlying mechanism requires further investigation.

A cholesteric order in CAB films has previously been reported to be observed with transmission electron microscopy [13,25]. In the present study, a cholesteric-like pattern was also observed by SEM analysis for some of the CAB films. This possible cholesteric ordered phase was displayed in the SEM picture of a cross-section of the CAB-500-5 film, visible as a regularly repeating pattern of striation lines in Figure 6. Cholesteric ordering occurs with the polymer chains arranged in a twisted, long-range chiral order, and the phase layers become incrementally displaced in helical stacking. When the film is cut, the knife alternatively cuts parallel or perpendicular to the direction of the polymer chains; so, the striation lines will be caused by the chiral pitch. It is suggested in the literature that CAB films can display this type of phase, but the impact on the phase obtained from different CAB films is not systematically reported [13]. In this study, we have found that, while some samples apparently show signs of possible cholesteric order, such as CAB-500-5, CAB-381-0.1, and CAB-381-2, others such as CAB-551-0.01 and CAB-381-20 do not exhibit this clearly. Therefore, we conclude that CAB with a molecular weight as low as 12.2 kDa and a DS_Bu_ as low as 1.76 could possibly show a cholesteric phase. In Figure 6, an SEM image showing the cross-section of the CAB film samples illustrates the difference in order. The number of hydroxyl groups in the polymer chains, together with the DP, constitutes a distinction between the CAB films that show signs of cholesteric phase formation and those that do not. In CAB films with a higher number of OH groups, more hydrogen bonding is induced and less cholesteric order occurs since the hydrogen bonding interrupts the helix alignment. However, the CAB-381 series showed that the hydroxyl groups are not the only factor; with the same amount of OH, a difference was seen for low-viscosity samples that showed cholesteric ordering and high-viscosity samples that did not. The circles in Figure 6 show where possible striation lines are; however, those are unconfirmed to be actual striation lines from a cholesteric phase and might be an artifact caused by something else. To truly confirm the cholesteric order in any material requires further study, with analysis such as UV/Vis and perhaps circular dichroism spectroscopy, as well as polarizing microscope images observations. While observations of cholesteric-like structures in CAB films have been found in this study and in a previous study by Davé and Glasser [26], the structure is, to the best of our knowledge, still unconfirmed.

The possible cholesteric ordering in CAB reported in the literature shows a periodicity of the helix pitch from 500 nm to 900 nm [26], which is significantly larger than for the most similar sample in this study, CAB-500-5, where the observed periodicity between the possible striation lines was approximately 160 nm. The study reported a butyryl content of 2.57 and a molar mass of 53 kDa, and similar data from the sample compared in the present study were 2.54 and 57 kDa, respectively. The study also reported a pitch of only 25 to 45 nm for cellulose triacetate (CTA) and explained the difference between CTA and CAB with the bulkiness of the side-chain substitutions, which may have prevented compact organization and, thus, led to a larger pitch [26]. A discussion on the pitch height was provided in another previous study, where the cholesteric ordering of a butyrate containing cellulose derivative butyric esters of (2-hydroxypropyl) cellulose (BuPC) is examined. It is shown in that study that the pitch height is correlated to the degree of substitution, where the pitch height decreases with an increasing value of DS for BuPC [27], directly going against the discussion in the CAB study. However, in this study, the bulkiness of the substituents is analogous with the CAB study—the side group bulkiness cannot clearly explain the difference in the observed periodicity. One probable explanation originates from the solvent “goodness” and the differences in the evaporating kinetics of the solvents used for casting in the film preparation. The solvent used in the previous study was dimethylacetamide (DMAc) with a reported evaporation time of 5 days, while in this study, acetone was used, resulting in an evaporation time of only 2 days. This points to slower evaporation kinetics (with time for energetically favorable molecular placement) appearing to increase the chiral twist layer thickness, resulting in a wider periodicity.

## 4. Conclusions

The hypothesis for this study was that a variation in the degree of substitution of butyrate and a variation in the molecular weight of commercially available cellulose acetate butyrate can affect the structural order and film formability of a solvent-cast film. Film formability is a subjective term that has been introduced in this paper, describing the combined effects of brittleness, adhesion to the cast mold, and homogeneity of the visual film appearance. Fewer hydroxyl groups along the backbone and a high molecular weight are attributed to increasing film formability, as the hydrogen bonding decreases homogeneity and an increase in viscosity decreases the degree of freedom of the molecules. Both XRD and TOPEM DSC point to the material being semi-crystalline. The crystallinity in the CAB films is affected by the choice of side groups, and their number and position along the backbone, especially the hydroxyl groups, that disturb the pattern of order. There is also more crystallinity with decreasing molecular weight, which is attributed to the increasing degree of freedom of the molecules. It is likely that all films experience cold crystallization, although this has only been proven for one form of CAB due to the likely overlap with the enthalpic relaxation peak, as both are detectable in the non-reversing heat flow signal at similar temperatures. While cholesteric ordering was previously suggested to occur in CAB films with a high butyrate content, it has now been shown that signs of cholesteric ordering in the CAB films can occur in a sample with a butyrate content as low as 1.76 and a molecular weight as low as 12.2 kDa. Hydrogen bonding has been identified as an interruption to the helix alignment of cholesteric ordering. Following this study, the knowledge gap regarding the understanding of the order in CAB films has become less substantial, although it is far from completely closed.

## Figures and Tables

**Figure 1 polymers-15-02205-f001:**
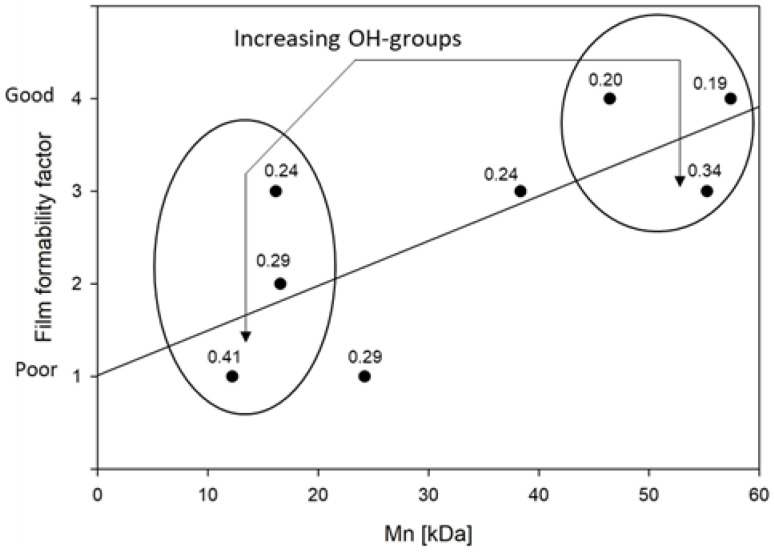
Film formability relates to both molecular weight and number of OH-groups. The DS*_OH_* is stated by each sample’s data point. The molecular weight is more influential on film formability than the DS*_OH_* as the former trend is only apparent during similar M_n_. A linear regression line is added to raise awareness to the overall trend.

**Figure 2 polymers-15-02205-f002:**
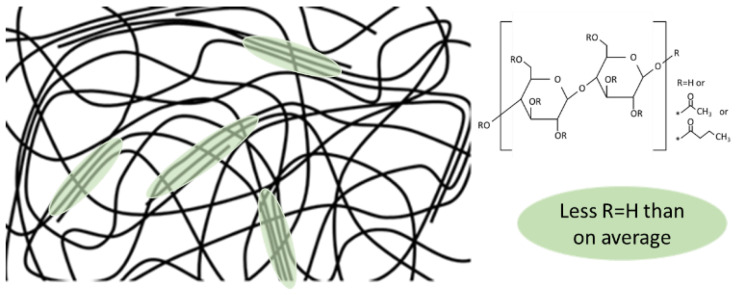
A suggested illustration of semi-crystalline order in the films, the * marks where R is substituted.

**Figure 3 polymers-15-02205-f003:**
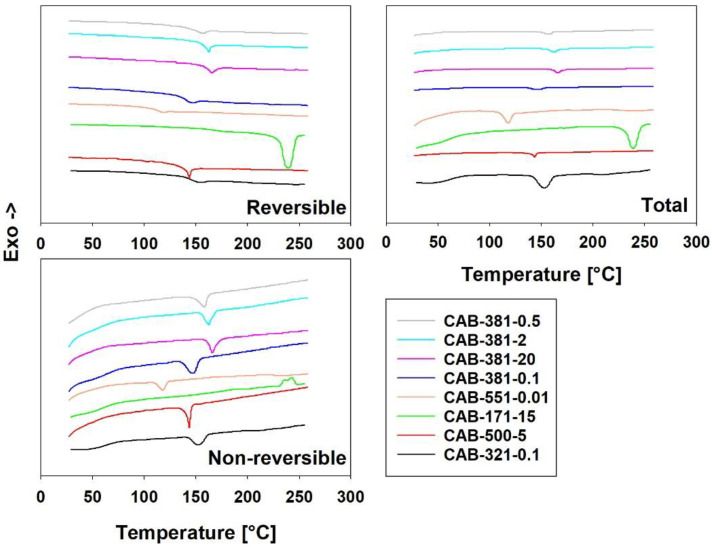
Modulated DSC curves. The reversible (**top left**) curves show most of the melting (endothermic). The non-reversible curves (**bottom left**) show enthalpic relaxation (endothermic), cold crystallization (exothermic), and some of the melting (endothermic), which all occur around the same temperature. The total (**right**) curves are a combination of the reversible and non-reversible curves and equivalent with unmodulated DSC.

**Figure 4 polymers-15-02205-f004:**
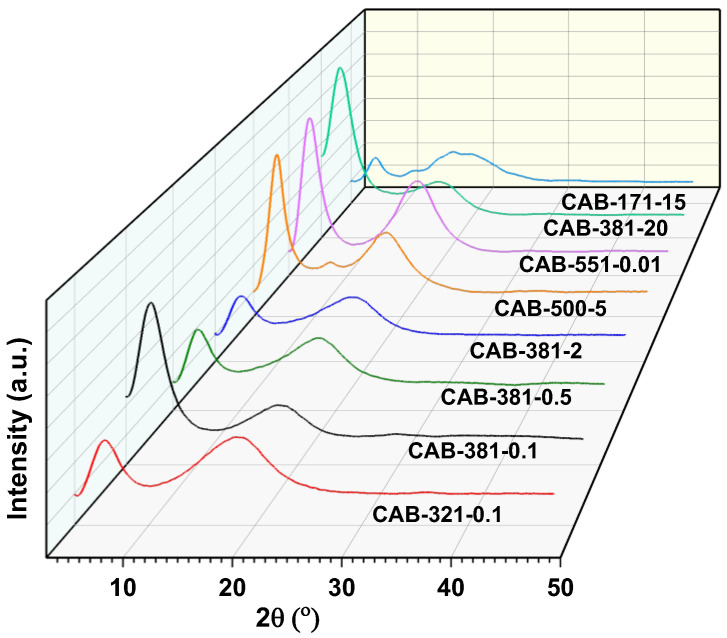
XRD patterns for the CAB films. Subgrouping with constant DS_Bu_ and M_n_ shows a trend with increasing first-peak intensity with increasing DS_Bu_ and decreasing molecular weight.

**Figure 5 polymers-15-02205-f005:**
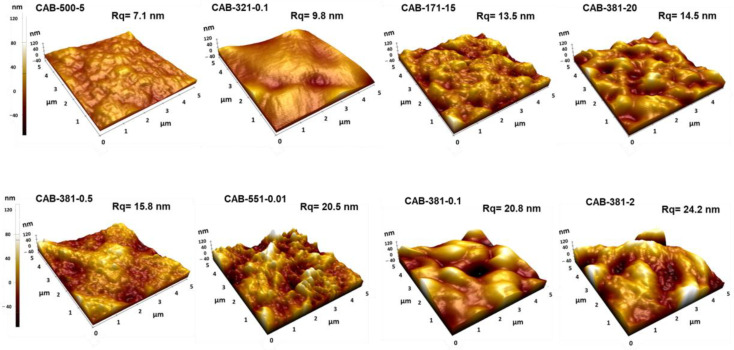
AFM 3D height image and surface roughness of films prepared from various CAB samples.

**Figure 6 polymers-15-02205-f006:**
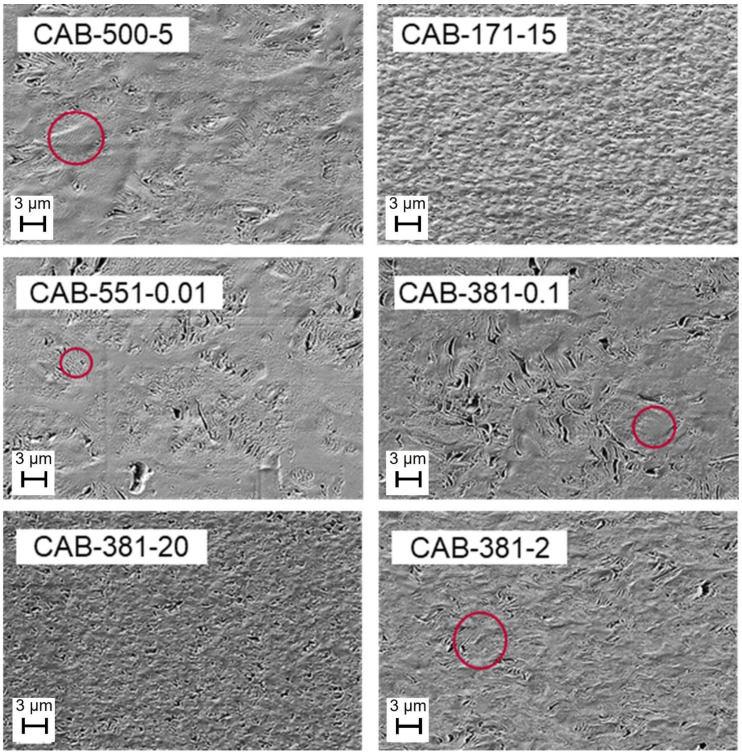
SEM pictures showing CAB film cross-sections. CAB-500-5, CAB-551-0.01, CAB-381-0.1, and CAB-381-2 show signs of possible cholesteric ordering (examples are given in red circles), while CAB-171-15 and CAB-381-20 show little to no observable examples of cholesteric ordering.

**Table 1 polymers-15-02205-t001:** CAB supplier data [4].

Sample/Eastman Name	Acetyl (wt%)	Butyryl (wt%)	Hydroxyl (wt%)	Viscosity (Sec)
CAB-321-0.1	17.5	32	1.3	0.1
CAB-500-5	3.0	51	1.0	5
CAB-171-15	29.0	18	1.1	19
CAB-551-0.01	2.0	52	2.0	0.02
CAB-381-0.1	13.5	38	1.5	0.1
CAB-381-20	13.5	37	1.8	20
CAB-381-2	13.5	38	1.3	2
CAB-381-0.5	13.5	38	1.5	0.5

**Table 2 polymers-15-02205-t002:** Molecular weight from the SEC analysis and molar-mass dispersity.

Sample	M_n_ (kDa)	M_w_ (kDa)	M_z_ (kDa)	MMD
CAB-321-0.1	16.2	24.0	35.3	1.5
CAB-500-5	46.5	88.3	145.7	1.9
CAB-171-15	57.4	104.8	177.0	1.8
CAB-551-0.01	12.2	17.8	25.8	1.5
CAB-381-0.1	16.6	24.5	36.3	1.5
CAB-381-20	55.3	113.7	194	2.1
CAB-381-2	38.4	66.7	108.3	1.7
CAB-381-0.5	24.2	41.9	67.7	1.7

**Table 3 polymers-15-02205-t003:** Calculated values for the degree of substitution, molecular weight of one average repeat unit, and degree of polymerization.

Sample	DS_OH_	DS_Ac_	DS_Bu_	M_unit_	DP
CAB-321-0.1	0.24	1.28	1.42	316	76
CAB-500-5	0.20	0.24	2.48	346	255
CAB-171-15	0.19	2.03	0.76	301	348
CAB-551-0.01	0.41	0.16	2.54	347	51
CAB-381-0.1	0.29	1.03	1.76	329	75
CAB-381-20	0.34	1.01	1.68	322	353
CAB-381-2	0.25	1.03	1.76	329	203
CAB-381-0.5	0.29	1.03	1.76	329	127

## Data Availability

The data presented in this study are available in the article and Appendix A.

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
