# Peer review of "On Structural and Molecular Order in Cellulose Acetate Butyrate Films"

_polymers, 2023, doi:10.3390/polym15092205_

Round 1

Reviewer 1 Report

The TOPEM DSC data are interesting. However, this paper seems to be unable to assert anything because there are vague points about the pretreatment of the samples and the premise of thermal transition. In order not to further confuse the interpretation of the thermal analysis data of cellulose derivatives, which is already confused, unfortunately, publication of this paper with the current argument should not be considered.

A reconsideration of the publication would require a fundamental change in the interpretation of the thermal transition. Specifically, the main problems are as follows.

Cholestericity: Certainly, Glasser et al. may have stated that CAB has cholesteric properties, but it is highly questionable. There is no evidence of cholesteric structure in this paper. To determine the presence of cholesteric structure, we need careful discussion not only by polarizing microscope images (observation of spontaneous formation of fingerprint-like texture, band structure after shear application, and darkening of the field of view in the liquid crystal-isotropic phase transition at elevated temperature) but also by UV/Vis and CD spectroscopy. Without them, the publication value of this paper itself cannot be measured.

Caption for Figure 4: XRD is not spectroscopy.

Thermal history: The authors seem to be indifferent to the thermal history of samples subjected to XRD and DSC. It seems that as-prepared films were used for measurement in XRD, but comparison should be made between films that have undergone heat treatment such as annealing and have a uniform thermal history. In DSC, it is reasonable to adopt the second heating data, but there is no information about the cooling rate. The timescale of observation is very important.

Reviewer 2 Report

Abstract

·        The abstract did not contain the valuable gain results. So, the abstract must be rewritten

Preparation method

·        The used materials must be supported by the company nale the country that purchased from

Result and discussion

·        In XRD diffractogram, it was appeared number and different diffraction peaks, where is the indices of these peaks and the reference XRD card number that used to illustrate these diffractogram?

·        It was better to take a look of the elemental analysis of the prepared samples, at least one sample, by EDX

·        Figure 2 could be removed or enhanced to be more valuable for readers

·        Also, it is cellulose acetate butyrate, which represent a hdrocarabon materials. So, where is the spectroscopic investion technique to support your result like FT-IR

Reviewer 3 Report

Reference paper: polymers-2172164

I have read the article entitled “On structural and molecular order in cellulose acetate butyrate Films” and my comments are summarized below.

Major comment: The manuscript elucidated the molecular structure order in cellulose acetate butyrate films using several analytical tools. Different characterizations have been performed to determine the required important parameters. The paper is of interest to many researchers. The work has apparently been well executed. However, additional information and modifications are necessary to improve the scientific content of the article. The paper could be published with major revisions.

1– Language should be thoroughly revised as some of the sentences are confusing and some errors can be found.

2–The originality of the paper needs to be further clarified in the introduction part. Several references are very old. Highlight the motivation to understand the evolution of the structure of the cellulosic derivative. I request to use some recent references dealing with cellulose, its derivatives, and its application such as:1

3–I miss some important references dealing with cellulose, its derivatives, and its application. Add some recent references to the introduction part to improve the content, such as .

https://github.jbcj.top:443/https/doi.org/10.1016/j.chemosphere.2022.133914

https://github.jbcj.top:443/https/doi.org/10.1039/D2NR01967A

https://github.jbcj.top:443/https/doi.org/10.1002/9783527827589.ch8

4– Provide references for Equations 1-6. Can they be applied to other cellulose derivatives? What are the main input and what is the source of these parameters?

5– A schematic presentation of the procedure used to produce the film is highly recommended.

6– Line 131: use 'crucible" instead of "cubicle". What is the atmosphere used and its flow rate? How about the calibration of your DSC?

7– Page 7:The kinetic is considered as the non-reversing signal (f(T,t)), while the reversing signal is dependent on the specific heat. The enthalpic relaxation is different from the melting process. Correct and improve the discussion of this part. A high-quality presentation of Figure 3 is requested.

8–Conclusions can be improved accordingly.

Round 2

Reviewer 2 Report

The paper can be accepted in the current form

Reviewer 3 Report

The authors have revised the manuscript according to the reviewer comments, and the paper can be acceptable for publication in its present form.